

# Contactless optical hygrometry in LACIS-T

Jakub L. Nowak[1,*], Robert Grosz[1,*], Wiebke Frey[2], Dennis Niedermeier[2], Jędrzej Mijas[3], Szymon P. Malinowski[1], Linda Ort[2,4], Silvio Schmalfuß[2], Frank Stratmann[2], Jens Voigtländer[2], and Tadeusz Stacewicz[3]

[1]Institute of Geophysics, Faculty of Physics, University of Warsaw, Pasteura 5, 02-293 Warsaw, Poland
[2]Experimental Aerosol and Cloud Microphysics, Leibniz Institute for Tropospheric Research, Permoserstr. 15, 04318 Leipzig, Germany
[3]Institute of Experimental Physics, Faculty of Physics, University of Warsaw, Pasteura 5, 02-293 Warsaw, Poland
[4]now at Alfred Wegener Institute for Polar and Marine Research, Am Handelshafen 12, 27570 Bremerhaven, Germany
[*]These authors contributed equally to this work.

**Correspondence:** Tadeusz Stacewicz (tadeusz.stacewicz@fuw.edu.pl)

**Abstract.**

The Fast Infrared Hygrometer (FIRH), employing open-path tunable diode laser absorption spectroscopy at the wavelengths near 1364.6896 nm line, was adapted to perform contactless humidity measurements at the Turbulent Leipzig Aerosol Cloud Interaction Simulator (LACIS-T), a unique turbulent moist-air wind tunnel. The configuration of the setup allows for scanning

at various positions without the need for repeated optics adjustments. We identified three factors which significantly influence the measurement – self-broadening of the absorption line, interference in the glass windows and parasitic absorption in the ambient air outside the tunnel – and developed correction methods which satisfactorily account for these effects. The comparison between FIRH and a reference hygrometer (dew-point mirror MBW 973) indicated a good agreement within the expected errors across the wide range of water vapor concentration $1.0 \ldots 6.1$ cm$^{-3}$ (equivalent to dew-point temperature of

$-5.4 \ldots +21$ °C at the temperature of 23 °C).

High temporal resolution ($\sim 2$ kHz) allowed for studying turbulent fluctuations in the course of intensive mixing of two air streams which had the same mean velocity but differed in temperature and humidity, including also the settings for which the mixture can be supersaturated. The obtained results complement the previous characterizations of turbulent velocity and temperature fields in LACIS-T. The variance of water vapor concentration exhibits a maximum in the center of the mixing zone

which coincides with the steepest gradient.

## 1 Introduction

Water vapor is the component of the atmosphere which is of particular importance for shaping weather and climate. The efficient absorption of terrestrial radiation makes it the most potent greenhouse gas and its phase transitions result in the formation of clouds and precipitation as well as latent heat transport.

The distribution of water vapor in the atmosphere is highly inhomogeneous across the range of scales. At the largest scales, typical conditions differ from relatively moist atmospheric boundary layer to rather dry upper troposphere/lower stratosphere





and from moist tropics to dry polar regions. In addition, substantial gradients of humidity often occur at the surface and top of the boundary layer or at cloud edges (Matthews et al., 2014; Haman et al., 2007; Malinowski et al., 2013). At the smallest scales, turbulent fluctuations of humidity and temperature determine local supersaturation in which individual aerosol particles

can be activated to form cloud droplets or ice crystals and further grow through condensation or deposition (Chandrakar et al., 2016, 2017, 2018; Desai et al., 2018).

Highly accurate and high resolution (spatial or temporal) measurements of water vapor concentration, both in field and laboratory experiments, are increasingly demanded to address contemporary research questions regarding cloud microphysics and cloud–turbulence interactions. High accuracy is essential to investigate the nucleation and growth of ice crystals in ice and

mixed-phase clouds (Spichtinger et al., 2004; Peter et al., 2006; Krämer et al., 2009) whereas high resolution is crucial to obtain reliable statistics of local supersaturation which control stochastic condensation under turbulent conditions (Prabhakaran et al., 2020; Thomas et al., 2021).

Despite considerable progress in the development of hygrometers for airborne, ground-based and laboratory applications (e.g. May, 1998; Diskin et al., 2002; Podolske et al., 2003; Zondlo et al., 2010; Beaton and Spowart, 2012; Meyer et al., 2015;

Neis et al., 2015a, b; Thornberry et al., 2015; Metzger et al., 2016; Nowak et al., 2016; Buchholz et al., 2017; Stacewicz et al., 2018), the comparability between different instruments remains insufficient. Large discrepancies of up to 20% are observed even under controlled laboratory conditions (Fahey et al., 2014). Furthermore, the quality of humidity measurements often lags in accuracy and resolution behind the state-of-the-art measurement techniques relevant for other atmospheric parameters, e.g. temperature. As a result, the limitations of humidity measurements prevent an improved understanding of some important

physical processes. This fact can be illustrated by the examples from field and laboratory studies. In the observations of mixing at stratocumulus top performed by Siebert et al. (2021), see Fig. 14 therein, the small-scale features of the mixing process are clearly indicated by the temperature records but the same structures cannot be identified in the simultaneous humidity records due to insufficient resolution. Furthermore, the recent International Cloud Modeling Workshop considered the case of turbulent moist convection inside the Michigan Tech Pi Chamber (Chang et al., 2016) and revealed many differences between

the numerical models participating in the comparison (Chen and Krueger, 2021). It was concluded that each model exhibits different statistics of supersaturation (mean and variance) and it is highly desirable to know which values are relevant for the convection in the chamber. However, this cannot be discerned without appropriate accurate and high resolution measurements of humidity.

Similarly, Niedermeier et al. (2020) provided statistics of turbulent temperature fluctuations (see Fig. 6 therein) in the Turbu-

lent Leipzig Aerosol Cloud Interaction Simulator (LACIS-T), a unique turbulent moist-air wind tunnel designed to investigate the interactions between cloud microphysics and small-scale turbulence. However, with the available instrumentation they could not obtain analogous results for humidity fluctuations.

Within the present study, we adapted the Fast Infrared Hygrometer (FIRH), an instrument employing tunable diode laser absorption spectroscopy (Nowak et al., 2016), to perform humidity measurements at LACIS-T. The goal of the series of

experiments was two-fold: (1) to evaluate the properties of FIRH under a wide range of well-defined reproducible conditions





resembling those in the real atmosphere, (2) to characterize the humidity field and turbulent fluctuations of humidity inside LACIS-T for different settings of the tunnel.

LACIS-T is an ideal facility to test FIRH because temperature and humidity in each of the two streams entering the measurement volume can be precisely controlled, while the turbulent mixing of the streams produces fast fluctuations of temperature and humidity (Niedermeier et al., 2020). On the other hand, FIRH is well-suited to resolve small-scale and quickly changing features of the humidity field inside LACIS-T because it provides high temporal resolution and its typical optical path roughly corresponds to the width of the LACIS-T measurement section (Nowak et al., 2016). This enables contactless optical sampling from outside the tunnel which eliminates the influence of the instrument on the investigated processes. Such a need of a contactless sampling was recognized following the reports from other laboratory experiments (e.g. Anderson et al. (2021) observed that the position of sensor holders inside the Pi chamber affects the orientation of the principal circulation) and taking into account the relatively small size of the central section of LACIS-T.

The present paper is structured in the following way. Section 2 introduces the LACIS-T facility as well as the FIRH instrument and explains the adaptations applied to the hygrometer with respect to its original version. Section 3 outlines the specific physical factors which strongly influence the measurement and need to be corrected for in order to retrieve the true value of humidity: self-broadening of the absorption line, interference in the glass windows and parasitic absorption in the ambient air outside the tunnel. Section 4 evaluates the accuracy of FIRH employing two approaches: a priori and with respect to a slow-response reference hygrometer. Section 5 presents and interprets the results of the measurements of mean humidity and turbulent fluctuations in the course of mixing of two streams inside LACIS-T for various selected stream settings. Eventually, section 6 summarizes and discusses the findings.

## 2 Instrumentation

### 2.1 LACIS-T facility

LACIS-T is a unique turbulent moist-air wind tunnel established to study cloud physical processes and the interactions between cloud microphysics and turbulence under a wide range of well-defined reproducible conditions resembling warm, mixed-phase and cold clouds. The design and capabilities of LACIS-T were described in detail by Niedermeier et al. (2020).

The wind tunnel works in a closed loop. Two air streams with separately controlled temperatures, humidities and velocities between 0.5 and 2 $\mathrm{m\,s^{-1}}$ are turbulently mixed inside the measurement section. For the current study, a fixed velocity of 1.5 $\mathrm{m\,s^{-1}}$ was used. The turbulence is generated by the passive square-mesh grids. Aerosol seeding can be additionally applied by isokinetically injecting aerosol particles directly into the mixing zone. The mixing of the two streams can be observed in the measurement section with the dimensions 80 x 20 x 200 cm (Fig. 1) through the windows of borosilicate glass. The measurement section is surrounded by a construction of rails (RK Rose & Krieger GmbH) allowing for the installation of various measurement apparatus and its displacement to selected positions.

LACIS-T is equipped with a set of instruments for aerosol particle generation, cloud particle sizing and monitoring the flow and thermodynamic conditions (Niedermeier et al., 2020, Table 1). In this study, we employed the dew-point mirror (DPM,

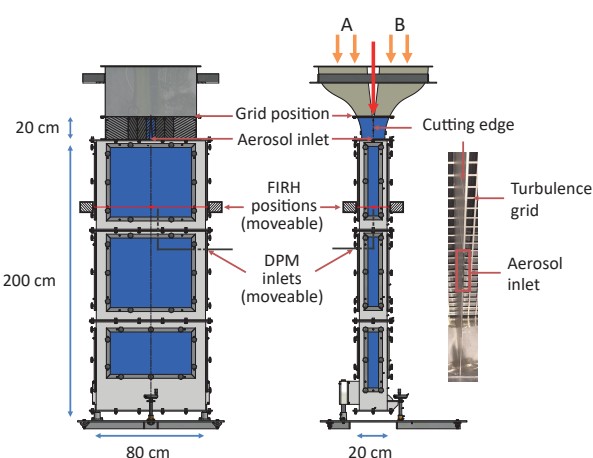

**Figure 1.** Schematic of the measurement section of LACIS-T. A and B mark the two air streams which are mixed in the measurement section. The red arrow marks the location where aerosol particles can be injected. Red line denotes the position of the FIRH optical path. Adapted from Niedermeier et al. (2020).

model 973 by MBW Calibration AG) as a slow-response reference hygrometer. It allows for the measurements of dew/frost-point temperature $T_d$ in the range of $-50\ldots+20\,°C$ with accuracy of $\leq \pm 0.1\,°C$ and reproducibility of $\leq \pm 0.05\,°C$ as well as temperature $T$ in the range of $-50\ldots+100\,°C$ with accuracy of $\leq \pm 0.07\,°C$ and reproducibility of $\leq \pm 0.04\,°C$ at the rate of 1 Hz. The air was sampled with a steel inlet head facing the flow and positioned below ($\sim 1$ cm) the horizontally oriented optical path of FIRH (see sec. 2.2). Although there is a possibility of developing an upstream disturbance of the flow due to the inlet, the influence on FIRH measurements is expected to be negligible due to the small size of the inlet (diameter of 6 mm) in relation to the length of the FIRH optical path. The air in the laboratory outside the tunnel is dried by a dedicated conditioning system to about $T_d = -10\,°C$. The ambient conditions are monitored with digital sensors (Si7021 and MPL3115A2) capable of measuring $T$ and $T_d$ with accuracy of $\pm 0.4\,°C$ and $\pm 0.8\,°C$, respectively. With those values, one can easily calculate water vapor concentration $n$ according to:

$$n = \frac{e_s(T_d)N_A}{RT} \tag{1}$$

where $e_s$ is saturation vapor pressure, $N_A$ denotes the Avogadro and $R$ the universal gas constant. The dependence of $e_s$ on temperature results from the Clausius-Clapeyron relation. In the numerical calculations involved in this study, we employed the polynomial approximations given by Flatau et al. (1992).



## 2.2 FIRH instrument

FIRH is an open-path optical sensor developed for quick measurements of small-scale humidity fluctuations in turbulent atmo-
spheric flows. The design, operation, properties and comparison of this instrument with selected other meteorological hygrom-
eters were described in detail by Nowak et al. (2016).

The basic measurement principle is the quenching of infrared laser light whose wavelength is precisely tuned to a specific
absorption line of $H_2O$ molecule. In fact, the attenuation at two different wavelengths $\lambda_M$, $\lambda_R$ corresponding to the neighboring
maximum $\sigma_M$ and minimum $\sigma_R$ of the absorption cross section is compared. For such close wavelengths, the absorption by
glass optical elements, scattering by dust or water droplets as well as sensitivity of detectors are practically the same while
the difference in absorption by water vapor molecules is substantial (see absorption spectrum in Fig. 2). The exact tuning of
the wavelength prevents any interferences by other absorbing compounds present in the atmosphere, e.g. $CO_2$. Therefore, such
differential measurement is sensitive only to the mean concentration of water vapor molecules $n$ along the optical path of length
$L$ between the emitter and the detector. This concentration can be determined with the equation resulting from Lambert-Beer
law:

$$n = \frac{1}{(\sigma_M - \sigma_R) L} \ln\left(\frac{\mathcal{I}_1(\lambda_M)}{\mathcal{I}_2(\lambda_M)} \frac{\mathcal{I}_2(\lambda_R)}{\mathcal{I}_1(\lambda_R)}\right). \tag{2}$$

where $\mathcal{I}_1$ and $\mathcal{I}_2$ are the intensities of the light beam entering and leaving the sampled volume, respectively. The concentration
can be converted into other humidity units (e.g. water vapor partial pressure $e$, specific humidity $q$ or $T_d$) with standard
thermodynamic formulas.

The same absorption line as in the earlier version of FIRH was used: $\lambda_M = 1364.6896$ nm. However, a different reference
wavelength $\lambda_R = 1364.8371$ nm was selected in order to ease the frequent switching between the two wavelengths which can
then be achieved by changing the laser current only while keeping the laser temperature fixed. According to the HITRAN
database of absorption spectra (Rothman et al., 2013), for relatively dry atmospheric conditions ($p = 1000$ hPa, $T = 23$ °C,
$n = 10^{16}$ cm$^{-3}$), the respective absorption cross sections equal $\sigma_M = 6.56 \cdot 10^{-20}$ cm$^2$ and $\sigma_R = 1.54 \cdot 10^{-21}$ cm$^2$ (see Fig. 2).

The implementation of FIRH in LACIS-T is schematically presented in Fig. 3. A single mode semiconductor laser (DL100,
Toptica Photonics AG) serves as a source of monochromatic light of a desired wavelength. Precise tuning of the laser to $\lambda_M$
or $\lambda_R$ is achieved with temperature and current controllers. The laser beam is conducted with a fiber and splits twice in the
couplers (10202A-90-APC, Thorlabs). Coupler 1 directs a portion of the beam (about 10% in intensity) into the wavelength
meter (WS6-200, HighFinesse GmbH) which is used instead of a high humidity reference cell applied by Nowak et al. (2016).
Feedback current signal from this instrument stabilizes the laser wavelength with the accuracy of $\leq 0.001$ nm and the precision
of $\leq 0.0001$ nm.

The main beam leaving the coupler 1 is sent to an electrooptic amplitude modulator (AM1550, JENOPTIK Optical Systems
GmbH) driven by a waveform generator (Handyscope HS5, TiePie engineering). Coupler 2 sends a portion of the beam (about
10 % in intensity) to the photodetector PD1 (FGA21, Thorlabs) that monitors the laser power. The dominant beam is further
guided to an emitter that directs it to the measurement volume. The intensity of the light transmitted through the sample is
measured with another photodetector (PD2) of the same kind at the opposite side of the tunnel. Signal digitization rate of





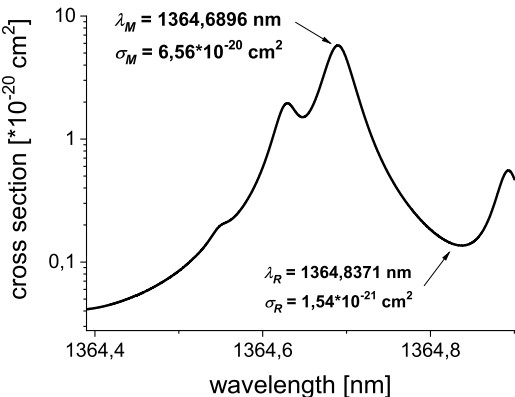

**Figure 2.** Absorption spectrum of the $H_2O$ molecule in the wavelength range relevant for the current study obtained using the HITRAN database ($p = 1000$ hPa, $T = 23\,^\circ$C, $n = 10^{16}$ cm$^{-3}$). The marked extrema are the wavelengths selected for laser tuning.

2 MHz was applied using a two channel 16 bit AD converter (Handyscope HS5, TiePie engineering) connected to a computer. The custom-developed software yields the final data rate of 2 kHz and handles two alternative methods of signal acquisition: (1) numerical lock-in demodulation if the amplitude modulator is active or (2) averaging of the direct high rate records if the modulator is deactivated. Coupler 2 is also used to merge an auxiliary small power 532 nm beam into the fiber. This beam is used only for system adjustments and not during the measurement.

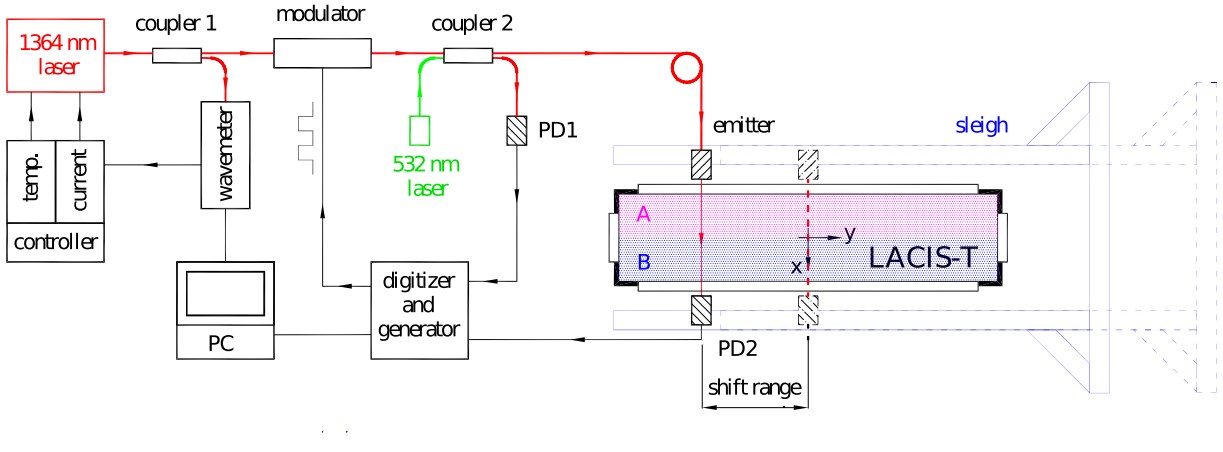

**Figure 3.** Schematic of the FIRH implementation at LACIS-T. The emitter and the photodetector PD2 are mounted on the movable sleigh which allows for the convenient scanning of the measurement volume: along fixed $x$ positions (long path, perpendicular to what is depicted in this scheme) or along fixed $y$ positions (short path, marked in this scheme).





The sampling of the air inside LACIS-T was achieved across the glass windows at the height $39\ \mathrm{cm}$ downstream the place where the two streams merge (see Fig. 1). The emitter and the photodetector PD2 were mounted on a rigid aluminium sleigh at the opposite sides of the tunnel (see Fig. 3) as close to the glass windows as it was possible (while maintaining the flexibility
of easy changes of the scanning position) in order to minimize the optical path outside the tunnel. Nevertheless, even despite drying the ambient air in the laboratory, parasitic absorption could not be entirely avoided (see sec. 3.3). The sleigh enables scanning the spatial variability of humidity statistics by moving the sensor horizontally along the walls of the tunnel. Two separate sleighs were prepared to allow the scanning at both perpendicular orientations depicted in Fig. 3: along a list of fixed $x$ positions ($L_x = 80 \pm 0.3\ \mathrm{cm}$), i.e. perpendicular to what is shown in Fig. 3, or along a list of fixed $y$ positions ($L_y = $
$20 \pm 0.3\ \mathrm{cm}$), as shown in Fig. 3.

At each position the measurement is accomplished in two steps. The laser wavelength is tuned once to $\lambda_M$, once to $\lambda_R$, and data records are stored for each wavelength. Because Eq. (2) involves the ratio of four intensities and the electric signals generated by the photodetectors PD1 and PD2 feature voltages $I_1$, $I_2$ proportional to the incoming light intensities $\mathcal{I}_1$, $\mathcal{I}_2$ regardless of the wavelength, the recorded values $I_1(\lambda_M)$, $I_2(\lambda_M)$, $I_1(\lambda_R)$, $I_2(\lambda_R)$ can be directly inserted into the equation.
Mean values of the record at $\lambda_R$ are used in the case of $I_1(\lambda_R)$ and $I_2(\lambda_R)$ while the timeseries recorded at $\lambda_M$ are inserted in the case of $I_1(\lambda_M)$ and $I_2(\lambda_M)$ in order to obtain a relevant timeseries of $n$.

## 3    Factors influencing the measurement

### 3.1    Absorption line properties

The shapes of spectral lines are mainly determined by collisions of the absorbing molecules with air particles (Demtröder,
2003). In rough approximation, the line profiles are described by Voight functions; however they are still a matter of investigation (Lisak and Hodges, 2007; Lisak et al., 2009; Regalia et al., 2014; Conway et al., 2020). Their parameters enabling the calculation of spectra at various circumstances are summarized in databases such as HITRAN (Rothman et al., 2013). The shapes weakly depend on the air pressure and temperature within typical range of those parameters in the atmosphere. Stronger dependence occurs for water vapor concentration due to self-broadening (Stacewicz et al., 2018). For the conditions relevant
for our experiment, the variations of the line shape due to pressure and temperature changes can be considered negligible. However, water vapor concentration in LACIS-T can vary from $\sim 10^{16}\ \mathrm{cm}^{-3}$ to $\sim 10^{18}\ \mathrm{cm}^{-3}$. In such a broad range, self-broadening leads to the considerable changes of $\sigma_M$ and $\sigma_R$ which are illustrated in Fig. 4. Therefore, the correct determination of $n$ by means of Eq. (2) has to involve the proper representation of those relationships.

In the data evaluation, we apply the values of absorption cross section obtained with the use of the HITRAN database for
$p = 1000\ \mathrm{hPa}$, $T = 23\ °\mathrm{C}$ (which is HITRAN reference temperature) and various levels of water vapor concentration (see Fig. 4). Following Buchholz et al. (2017) and Wunderle et al. (2006), we assume the conservative estimation of 3.5 % as the accuracy of $\sigma$.

The dependencies $\sigma_M(n)$ and $\sigma_R(n)$ were parameterized with smooth functions. The accuracy of such parametrization with respect to the data points extracted from HITRAN is $<0.1$ %, hence its effect on the accuracy of $\sigma$ is negligible. The parame-



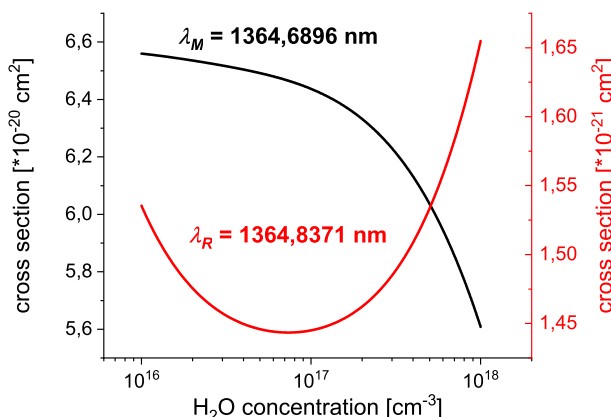

**Figure 4.** Dependence of $\sigma_M$ and $\sigma_R$ on water vapor concentration. Data extracted from the HITRAN database ($p = 1000$ hPa, $T = 23\,°$C).

terized functions $\sigma_M(n)$ and $\sigma_R(n)$ were used in Eq. (2) which then becomes an implicit relation to be solved numerically in order to calculate $n$.

### 3.2 Interference in the glass windows

The absorption spectrum of the glass is flat in the spectral range relevant for this study. Therefore, its influence on the measurement is negligible. However, multiple reflections of the light beam between the surfaces and the interference between the reflected beams lead to periodic oscillations in the transmission spectrum $\mathcal{T}(\lambda)$. For a single window, those fringes can be described by the formula (Demtröder, 2003):

$$\mathcal{T}(\lambda) = \left[1 + \mathcal{F}\sin^2\left(\frac{\delta}{2}\right)\right]^{-1} \tag{3}$$

where $\mathcal{F} = 4\mathcal{R}/(1-\mathcal{R})^2$ is the finesse coefficient and $\delta = 4\pi d\eta/\lambda + \Delta\varphi$ is the phase difference while $\mathcal{R}$, $\eta$ and $d$ denote the reflection coefficient, refractive index and thickness of the glass, respectively. If the incident light beam is perpendicular to the glass surface, then $\mathcal{R} = (1-\eta)^2/(1+\eta)^2$. Additional phase shift $\Delta\varphi$ follows from the uncertainty of the glass thickness.

For typical floated borosilicate 3.3 glass ($\eta = 1.47$) which was used in the LACIS-T windows one can evaluate that the surface reflection coefficient equals $\mathcal{R} = 3.6$ % and the finesse is about $\mathcal{F} = 0.16$. The transmission of a single window oscillates with $\lambda$ within quite a large range of $0.865 \leq \mathcal{T} \leq 1$ around the mean value of $\langle\mathcal{T}\rangle = (1-\mathcal{R})^2 = 0.929$. The period of the oscillation (i.e. wavelength difference between two neighboring maxima) can be estimated according to an approximate formula: $\Delta\lambda \approx \lambda^2/(2\eta d)$. The windows in LACIS-T are either 8 mm or 6 mm thick which results in the oscillation period of $\Delta\lambda = 0.08$ nm and $\Delta\lambda = 0.105$ nm, respectively.



In the case of two identical windows (marked $a$ and $b$) the effective transmission coefficient is equal to:

$$\mathcal{T}_2(\lambda) = \mathcal{T}_a(\lambda) \cdot \mathcal{T}_b(\lambda) = \left[1 + \mathcal{F}\sin^2\left(\frac{\delta_a}{2}\right)\right]^{-1} \cdot \left[1 + \mathcal{F}\sin^2\left(\frac{\delta_b}{2}\right)\right]^{-1} \tag{4}$$

where $\delta_a = 4\pi d\eta/\lambda + \Delta\varphi_a$ and $\delta_b = 4\pi d\eta/\lambda + \Delta\varphi_b$. The transmission $\mathcal{T}_2(\lambda)$ oscillates around the mean value $\langle\mathcal{T}_2\rangle = (1 - \mathcal{R})^4 = 0.864$. The oscillation period is the same as for a single window. However, the range of oscillations depends on the relative phase shift $\Delta\varphi_2 = \Delta\varphi_b - \Delta\varphi_a$. The largest range $0.748 \leq \mathcal{T}_2 \leq 1$ corresponds to $\Delta\varphi_2 = 0$. The examples of $\mathcal{T}_2(\lambda)$ for two different $\Delta\varphi_2$ are shown in Fig. 5.

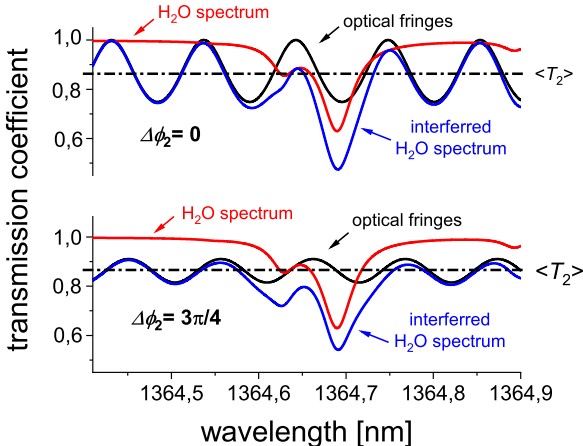

**Figure 5.** Examples of the optical interference in the glass windows ($d = 8\,\mathrm{mm}$) for two values of $\Delta\varphi_2$ which can occur for the transmission through LACIS-T (assumed $n = 10^{17}\,\mathrm{cm}^{-3}$).

Commonly, the described interference in the glass windows can be reduced with anti-reflection coatings applied on the glass surfaces or using thick or wedge optical windows. Exploiting Brewster angle of incidence and the light polarized parallel to the incidence plane also belong to the possible solutions. However, all these approaches were not applicable in the case of LACIS-T due to the size of the windows (tens of square decimetres in surface) and the desire to maintain their universal purpose.

In order to correct for the influence of the glass windows on the measurements with FIRH, we experimentally characterized this effect with a series of transmission scans. The tunnel flow was turned off and the windows lateral to the FIRH optical path were removed so that the thermodynamic conditions inside and outside the tunnel were the same. At each position $x$ or $y$ (see Fig. 3) used in the subsequent humidity measurements (see Table 1), the effective transmission coefficient $\mathcal{T}^{(g+l)}(\lambda)$ through the glass windows and the laboratory air was determined for the wavelengths in the range of $1364.46 - 1365.85\,\mathrm{nm}$. The wavelength was varied with the step of $0.001\,\mathrm{nm}$ by adjusting the settings of the wavemeter–laser controller stabilization loop. Analogous measurement was performed for the same path length but without the windows to obtain the transmission coefficient $\mathcal{T}^{(l)}(\lambda)$ through the laboratory air. The results corresponding to scanning along $x$ are presented in Fig. 6.



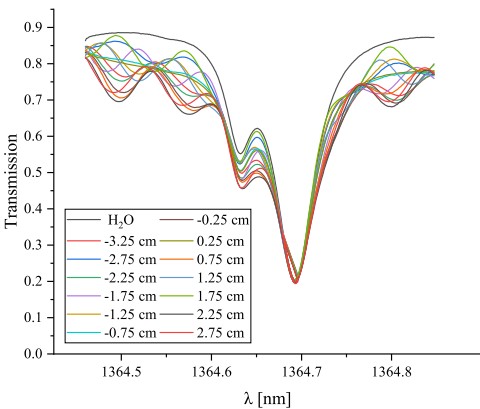

**Figure 6.** Effective transmission spectra $\mathcal{T}_x^{(g+l)}(\lambda)$ of the windows and laboratory air ($n_l = 2.93 \cdot 10^{17}$ cm$^{-3}$) at various positions $x$ compared with the transmission spectrum $\mathcal{T}_x^{(l)}(\lambda)$ of the laboratory air only. Node-like features are observed on both sides of the transmission minimum. One of them is located close to $\lambda_R$.

Periodic oscillations due to the interference in the windows can be observed in the entire investigated range. Their phases depend on the exact position $x$, probably due to the imperfections of the flatness of the glass surfaces and nonuniformity of the glass plate thickness. However, the curves exhibit node-like structure, i.e. the dependence of $\mathcal{T}^{(g+l)}$ on position becomes weak at some particular wavelengths. This is the case for $\lambda_M$ and $\lambda_R$. For this reason, we decided to neglect the dependence of the interference effect on the exact position. Such a simplification is reasonable taking into account the limited accuracy of

the position adjustment ($\pm 0.5$ mm). In the case of the perpendicular orientation, the node-like structure is not as clear but the amplitude of the oscillations is substantially smaller (not shown) which justifies the same approach.

The transmission due to the glass windows only can be estimated as the ratio $\mathcal{T}^{(g)} = \mathcal{T}^{(g+l)}/\mathcal{T}^{(l)}$. For the wavelengths exploited in FIRH, we derived $\mathcal{T}_x^{(g)}(\lambda_M) = 0.99 \pm 0.02$ and $\mathcal{T}_x^{(g)}(\lambda_R) = 0.87 \pm 0.01$ for scanning along $x$ while $\mathcal{T}_y^{(g)}(\lambda_M) = 0.99 \pm 0.01$ and $\mathcal{T}_y^{(g)}(\lambda_R) = 0.98 \pm 0.01$ for scanning along $y$. Those values can be applied as correction coefficients in order

to compensate the impact of window interference on humidity measurements. Hence, the measured PD2 signals $I_2$ involved in Eq. (2) were replaced with $I_2/\mathcal{T}^{(g)}$ to complete the correction.

### 3.3    Ambient conditions in the lab

As it was stated above, the emitter of the laser beam and the photodetector PD2 were mounted on the opposite sides of the tunnel in a way allowing for flexible scanning at different positions $x$ or $y$ without repeating laborious optical alignment (see

Fig. 3). Unfortunately, such a solution involves a portion of the optical path outside of the tunnel. The absorption over the total path of $L_l = 5.0 \pm 0.3$ cm in the laboratory air can be important in comparison with the absorption over the path $L$ inside, in particular for low humidity in the tunnel. Therefore, the conditions in the lab were monitored (see sec. 2.1) in order to account





for the effect of parasitic absorption by invoking the Lambert-Beer law. It can be estimated that the parasitic absorption in the laboratory exceeds 10 % of the absorption in the tunnel for about $T_d < -14.5\ ^\circ\mathrm{C}$ in the case of the scanning along $x$ and for

about $T_d < 3.7\ ^\circ\mathrm{C}$ in the case of scanning along $y$.

After including all the discussed corrections, the final formula for water vapor concentration in the tunnel takes the form:

$$n\left(\sigma_M(n) - \sigma_R(n)\right) = \frac{1}{L}\left[\ln\left(\frac{I_1(\lambda_M)\mathcal{T}^{(g)}(\lambda_M)}{I_2(\lambda_M)}\frac{I_2(\lambda_R)}{\mathcal{T}^{(g)}(\lambda_R)I_1(\lambda_R)}\right) - \left(\sigma_M(n_l) - \sigma_R(n_l)\right)n_l L_l\right] \tag{5}$$

where the terms on the right hand side are given by the measurements and the terms on the left hand side are functions of $n$ only. This equation is solved numerically to get $n$.

## 4    Characterization of FIRH

The accuracy of the measurement of water vapor concentration with FIRH was assessed with the two approaches: (1) a priori – by considering the maximum potential error introduced by the factors influencing the measurement (see sec. 3); (2) experimental – by comparing FIRH with a reference hygrometer (MBW973) under a range of conditions.

In the first approach, we neglected the inaccuracies related to the numerical solution of Eq. (5) and the parameterization

$\sigma(n)$. Those are expected to contribute negligibly in comparison with the errors related to other factors: $\sigma$, $L$, $I$, $\mathcal{T}^{(g)}$, $L_l$, $n_l$. Considering them, we derived an approximate formula for the expected maximum measurement error which involves two terms: relative and absolute errors. In the case of scanning along $x$, the relative error is $\sim 7.4\%$ and the absolute error is $\sim 8\cdot 10^{15}\ \mathrm{cm}^{-3}$. In the case of scanning along $y$, the relative error is $\sim 8.5\%$ and the absolute error is $\sim 2.3\cdot 10^{16}\ \mathrm{cm}^{-3}$. The dominant contribution to the relative error comes from $\sigma$ (followed by a smaller contribution of $L$) while the dominant

contribution to the absolute error results from $\mathcal{T}^{(g)}$ (followed by smaller contributions of $I$, $L_l$, $\sigma$ and $n_l$). Importantly, most of the observables ($\sigma$, $L$, $\mathcal{T}^{(g)}$, $L_l$, $n_l$) can be considered fixed during a single measurement series, yet known only with limited accuracy. As a consequence, the uncertainty of $n$ cannot be reduced by averaging many individual measurements. On the other hand, such systematic errors which are fixed over time do not affect derived turbulent fluctuations $n'$. Considering only the random error related to $I$, one would arrive at the absolute errors of $\sim 10^{15}\ \mathrm{cm}^{-3}$ and $\sim 3\cdot 10^{15}\ \mathrm{cm}^{-3}$ for scanning along $x$

and $y$, respectively.

In the second approach, we performed two comparison experiments consisting of a series of simultaneous measurements with FIRH and the dew-point mirror: at fixed $x$ (COMP-X) and at fixed $y$ (COMP-Y), see Table 1 and Fig. 3. The measurements were performed at various humidities inside the tunnel ($T_d = -21\ldots +21\ ^\circ\mathrm{C}$) while keeping the other LACIS-T parameters fixed, see Table 1. The thermodynamic conditions of the two streams were set the same ($T_A = T_B$, $T_{d_A} = T_{d_B}$) to avoid

the effects of mixing. For each humidity value, the records of 100 s were taken with the two instruments and their mean values served for the comparison. The results are presented in Fig. 7. For the dew-point mirror, water vapor concentration was calculated according to Eq. (1) which leads to the accuracy of $\leq 0.8\ \%$ based on the instrument specifications.





**Table 1.** List of experiments together with the corresponding wind tunnel and FIRH settings.

| Experiment | $T_A$ [°C] | $T_{d_A}$ [°C] | $n_A$ [$10^{17}$ cm$^{-3}$] | $T_B$ [°C] | $T_{d_B}$ [°C] | $n_B$ [$10^{17}$ cm$^{-3}$] | Position [cm] |
|---|---|---|---|---|---|---|---|
| COMP-X | 23 | -21…21 | 0.3…6.1 | $T_B = T_A$ | $T_{d_B} = T_{d_A}$ | $n_A = n_B$ | $x = 0.9$ |
| COMP-Y | 23 | -21…21 | 0.3…6.1 | $T_B = T_A$ | $T_{d_B} = T_{d_A}$ | $n_A = n_B$ | $y = 0.0$ |
| SCAN-X-1 | 23 | 20 | 5.7 | 23 | 10 | 3.0 | $x = -3.25…2.75$ |
| SCAN-X-2 | 23 | 20 | 5.7 | 23 | 4 | 2.0 | $x = -3.25…2.75$ |
| SCAN-X-3 | 20 | 20 | 5.8 | 4 | 4 | 2.1 | $x = -3.25…2.75$ |
| SCAN-X-4 | 22 | 20 | 5.7 | 12 | 10 | 3.1 | $x = -3.25…2.75$ |
| SCAN-Y-2 | 23 | 20 | 5.7 | 23 | 4 | 2.0 | $y = 0, -10, -20$ |
| SCAN-Y-3 | 20 | 20 | 5.8 | 4 | 4 | 2.1 | $y = 0, -10, -20$ |

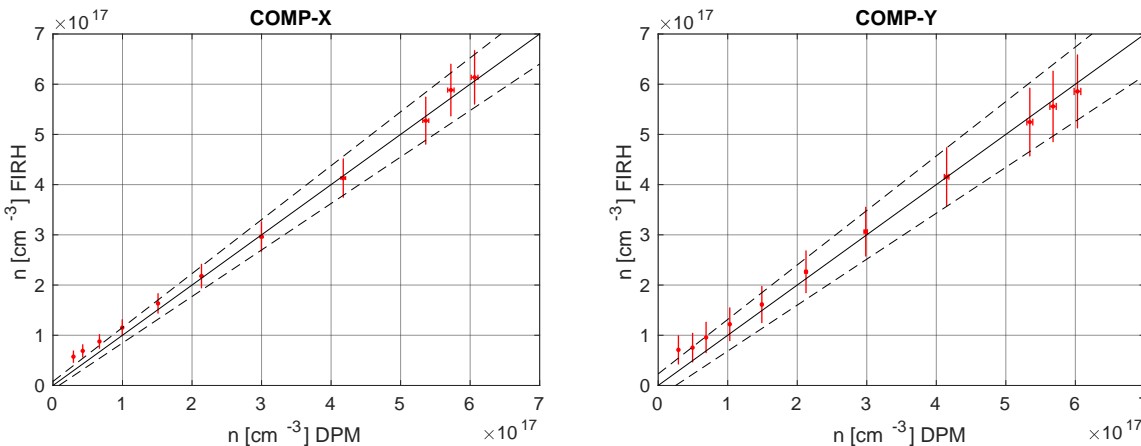

**Figure 7.** Comparison of FIRH with DPM (MBW973). Errorbars represents the estimated errors, solid black line is 1:1 ratio, dashed lines denotes the error range expected for FIRH (see text for details).

In general, the measurements with the two instruments agree with each other within the estimated error range across the most of the investigated humidity range. The dependence is highly linear (coefficient of determination $R^2 > 0.998$) for both data series. Root mean squared errors are $1.6 \cdot 10^{16}$ cm$^{-3}$ and $2.0 \cdot 10^{16}$ cm$^{-3}$ for COMP-X and COMP-Y, respectively.

At low humidity ($n < 10^{17}$ cm$^{-3}$, equivalent to $T_d < -5.4$ °C) the values of $n$ are overestimated by FIRH in comparison to DPM. For the case of very low humidity inside the tunnel, the three terms in the r.h.s. of Eq. (5) representing tunnel absorption $\ln\left(\frac{I_1(\lambda_M)}{I_2(\lambda_M)} \frac{I_2(\lambda_R)}{I_1(\lambda_R)}\right)$, window transmission $\ln\left(\frac{\mathcal{T}^{(g)}(\lambda_M)}{\mathcal{T}^{(g)}(\lambda_R)}\right)$ and ambient air absorption $(\sigma_M - \sigma_R)n_l L_l$ are of comparable magnitudes. Hence, the biases in the estimations of window transmission and ambient air absorption become particularly important



for the outcome. This effect is more pronounced for COMP-X than for COMP-Y due to the significantly higher (about 13 times) window transmission term (see also sec. 3.2).

## 5    Measurements of turbulent mixing inside LACIS-T

The mixing of the two air streams differing in thermodynamic properties was investigated in several measurement series named scans. Each scan consisted of a number of 300 s long records collected at various FIRH positions under fixed tunnel

settings given in Table 1. The DPM inlet was displaced in steps alongside the laser beam of FIRH so that the inlet was beneath the FIRH optical path. Two scans along $x$ (i.e. measurements at different $x$ positions, see sec. 2.2) explored the mixing of the streams under isothermal conditions ($T_A = T_B$) but different humidity (SCAN-X-1 and SCAN-X-2). Another two scans along $x$ investigated the mixing of the streams differing in both temperature and humidity (SCAN-X-3 and SCAN-X-4). The conditions in SCAN-X-3 allowed for creating a supersaturated mixture. Those four scans along $x$, each consisting of 13

positions, were followed by two scans along $y$ (i.e. measurements at different $y$ positions, see sec. 2.2), each consisting of 3 positions only, as no significant differences for the measurements with the laser beam averaging along the humidity gradient were expected. SCAN-Y-2 and SCAN-Y-3 were performed under the same settings of the tunnel as SCAN-X-2 and SCAN-X-3.

### 5.1    Mean conditions

The results of the scans along $x$ – mean $n$ and its variance – are presented in Fig. 8. The mean $n$ exhibits a significant

systematic offset (shift) between FIRH and DPM in all four experiments. This offset can be attributed to three factors: (1) the limited accuracy of FIRH (see sec. 4), (2) the displacement between the FIRH optical path and the DPM inlet (deliberate shift in vertical and inevitable inaccuracy in horizontal positioning), (3) inherent difference in sampling regime between the instruments (FIRH provides an average along the optical path while the DPM inlet collects air at a specific location). The offset is higher than observed in the comparison experiment COMP-X, likely due to the significant spatial gradient of humidity. Such

gradient was absent in COMP-X but here, due to the factors (2) and (3), it affects the outcome.

For scans along $y$, FIRH measurements represent an average along the optical path, hence along the humidity gradient, while DPM measurements are determined by the exact position of the inlet with respect to the mixing zone ($x = 0$ was targeted). Therefore, direct comparison of the measurement results is not justified.

In the course of SCAN-X-3, SCAN-X-4 and SCAN-Y-3, water vapor was observed to condense on the DPM inlet and cause

malfunctions of this instrument which explains irregularities in the DPM profiles in Fig. 8. This observation underlines an advantage of the contactless measurements with FIRH.

### 5.2    Turbulent fluctuations

High temporal resolution provided by FIRH allows to characterize not only the profile of the mean humidity across the measurement volume but also the properties of turbulent fluctuations in the course of mixing of the two streams. It should be noted,

however, that the measured fluctuations represent instantaneous, yet spatially averaged (along the optical path) humidities.





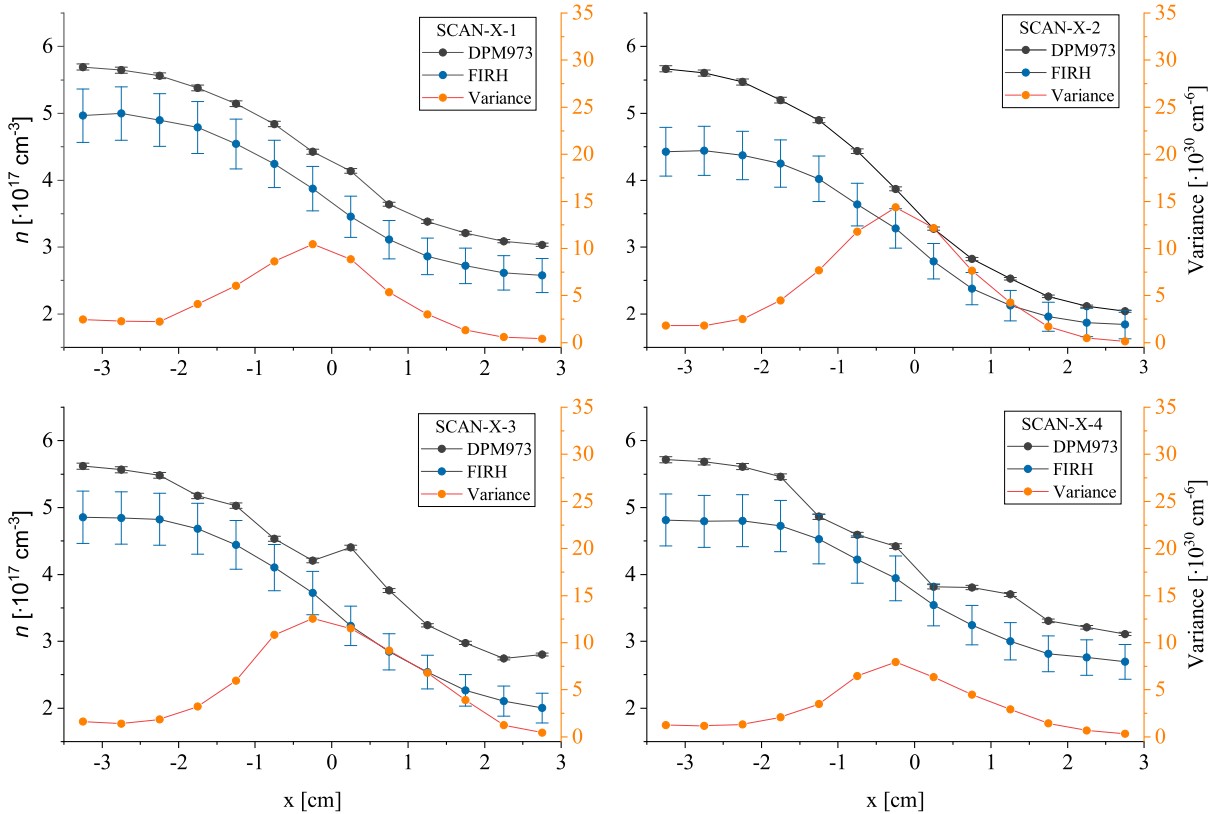

**Figure 8.** Turbulent mixing of the two streams differing in thermodynamic properties (given in Table 1) observed in the course of the four experiments: the profiles of mean $n$ and its variance with respect to the position $x$.

The profiles of $n$ variance are shown in Fig. 8. As expected, the variance is highest in the central part of the tunnel. Maximum variance coincides with the steepest gradient of the mean humidity. Variance reaches higher values for the experiments with a larger difference in $n$ between the streams (i.e. SCAN-X-2 and SCAN-X-3, see Table 1). Based on the variance profile, the width of the turbulent mixing zone at the height of our measurement is ∼5 cm, in agreement with Fig. 6 in Niedermeier et al.
300   (2020).

Recorded humidity fluctuations were further analyzed with the use of autocorrelation functions (ACFs) and power spectral densities (PSDs) derived for individual timeseries $n(t)$. ACFs for the four experiments from SCAN-X-1 to SCAN-X-4 are given in Fig. 9. The plots clearly indicate the dependence of the fluctuations on the position in the tunnel. Close to the center ($|x| < 1.5$ cm), ACFs decrease rapidly to cross zero at ∼0.018 s, reach maximum negative autocorrelation at ∼0.03 s and vary
around zero for larger time lags, suggesting the presence of oscillations in the flow which are coherent enough along the $y$ direction to be detected in longitudinally averaged signals. The oscillations are weak and vanish at a distance from the central plane ($|x| > 2$ cm). Outside the central part, ACFs decrease slower, almost monotonically, and reach zero at ∼0.25 s. This





behavior is subject to some variability with respect to the experiment, the side of the tunnel and the distance from the central plane.

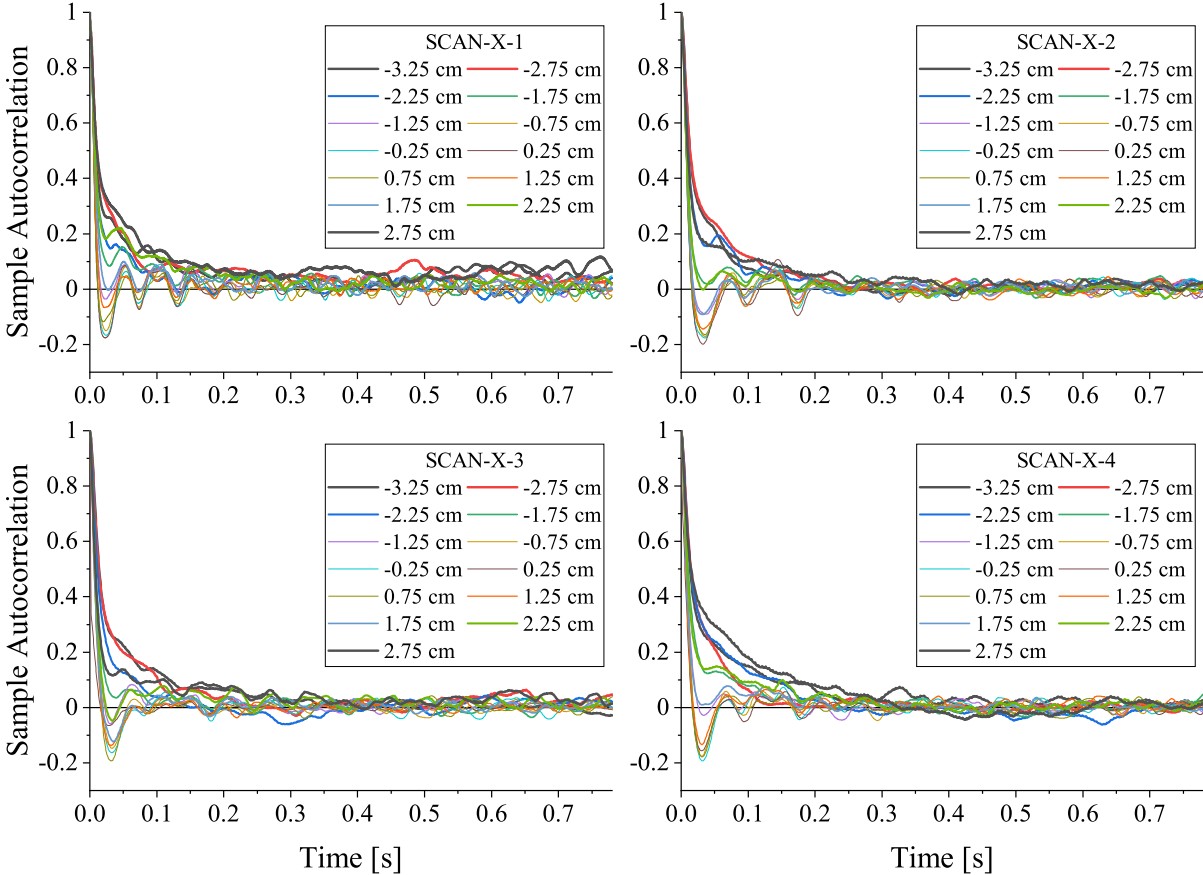

**Figure 9.** Autocorrelation functions of the timeseries $n(t)$ recorded at various positions $x$ during the four scans differing in the thermodynamic properties of the input streams (see Table 1 for tunnel settings). Thinner lines correspond to the positions close to the center while thicker lines represent the distant ones.

PSDs of the same timeseries are presented in Fig. 10. Close to the center ($|x| < 1.5$ cm), the PSDs exhibit a maximum at $\sim$14 Hz which is more pronounced in the case of isothermal conditions (SCAN-X-1 and SCAN-X-2) in comparison with non-isothermal ones (SCAN-X-3 and SCAN-X-4) which stays in accordance to the more regular fluctuations in the corresponding ACFs. Assuming Taylor frozen flow hypothesis and using the mean flow velocity $1.5 \, \mathrm{m\,s^{-1}}$, this frequency corresponds to the wavelength of $\sim$11 cm.

The characteristic frequency of $\sim$14 Hz identified in the signals might be related either to the effect of humidity changes inside air volumes or to flow velocity variations. We suppose the latter is more likely because when the aerosol flow in between the two streams is disabled (which is the case for our study), the profile of mean velocity in the central part of the tunnel





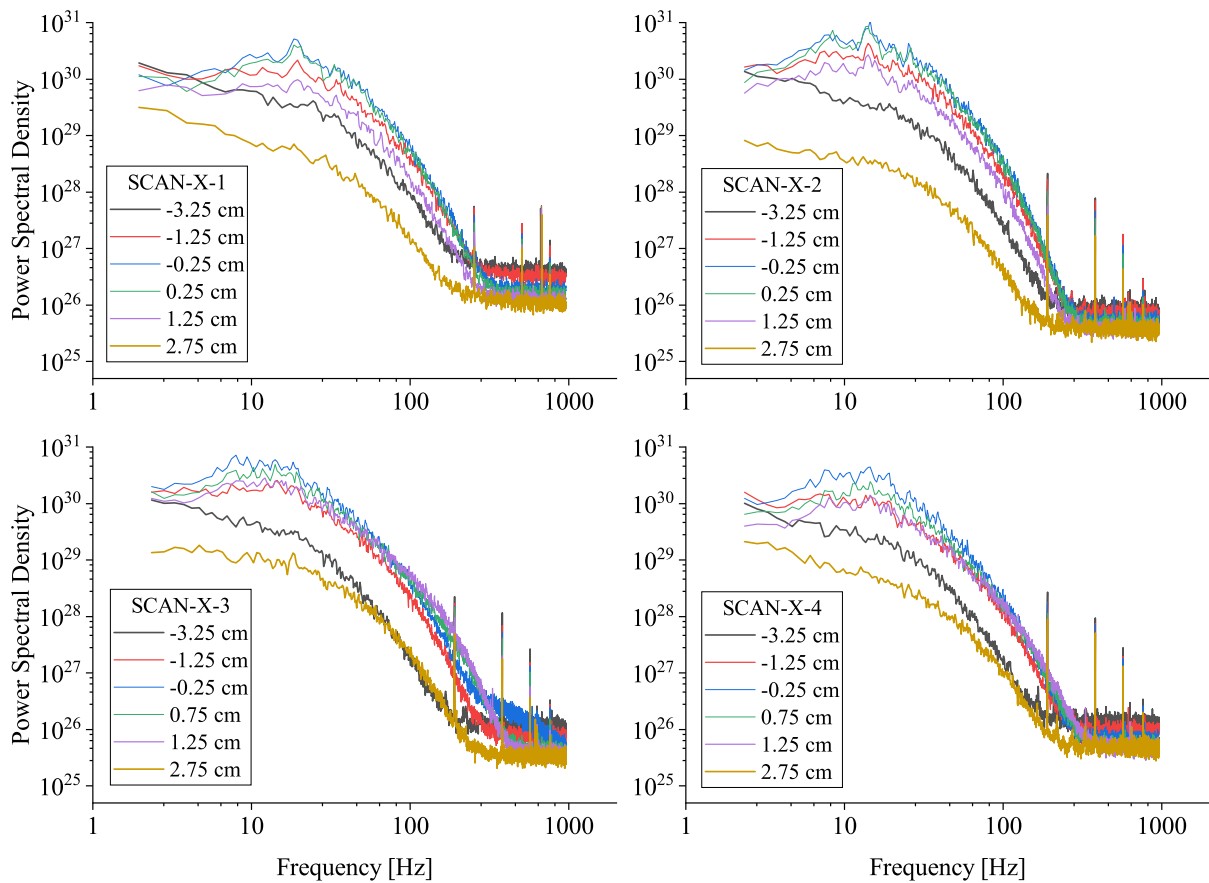

**Figure 10.** Power spectral densities of the selected timeseries $n(t)$ recorded at various positions $x$ during the four scans.

becomes inhomogeneous, see sec. 4.1. in Niedermeier et al. (2020). The spatial extent of this inhomogeneity presented there is $\leq 4$ cm (along $x$ direction). In order to investigate how such local mean velocity gradients in the central part affect the statistics averaged across the entire width, a separate experiment needs to be designed which would then explain the mechanism responsible for the observed PSDs and ACFs. But most importantly, the cloud formation studies at LACIS-T are unaffected by the mean velocity gradients due to the aerosol inlet because the configuration for cloud measurements involves enabled aerosol flow which provides homogeneous mean velocity profile in the central part (Niedermeier et al., 2020).

The results of SCAN-Y-3 are given in Fig. 11 and Fig. 12. They are similar to SCAN-Y-2 which is therefore not shown here. As noted before, the fluctuations recorded for this orientation are difficult to interpret due to the effective averaging along the humidity gradient. At $y = 0$ cm and $y = -10$ cm, the ACFs and PSDs indicate a significant contribution of the mode of the frequency of $\sim 47$ Hz and several others of higher frequencies. This mode is the strongest for $y = -10$ cm while the further ones (e.g. at $\sim 111$ Hz) are the strongest in the case of $y = 0$ cm. The observed complicated spectra might result from the combination of two effects. First, during the experiments, an additional inlet tubing for a second DPM (also MBW





973) was installed close behind the turbulence grid, at the position of $y = -5$ cm. This tubing, being right in-between the two measurement positions y = 0 cm and -10 cm, most likely caused flow disturbances which in the environment of strong gradient lead to increased humidity fluctuations. This is an important finding, so the tubing will be removed in future studies to avoid its influence on the flow. On the other hand, the minor vibrations of the windows (either 339 x 1148 x 6 mm or 584 x 1148 x 8 mm were used) can affect in a complicated manner the instantaneous net transmission $\mathcal{T}^{(g)}$ discussed in sec. 3.2. Because we do

not consider such transient effects in our correction method, the oscillations of the windows can influence the signal recorded by FIRH by a minor extent. Yet, it is unlikely that those oscillations change the humidity patterns inside the chamber.

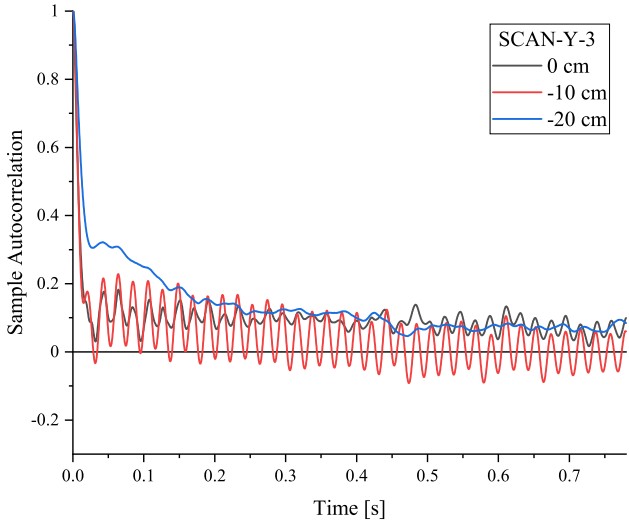

**Figure 11.** Autocorrelation functions of the timeseries $n(t)$ recorded at various positions $y$ during SCAN-Y-3.

At the frequency of $\sim$150 Hz, the PSDs reach the noise floor. For the scans along $x$, floor level slightly increases with increasing mean humidity along the beam, probably due to the combination of two effects: a decrease of the mean signal at the photodetector with increasing mean humidity (stronger absorption along the path) and different influence on the signal of dry

intrusion into humid environment at small $x$ (small change in total absorption) versus humid intrusion into dry environment at large $x$ (significant change in total absorption). At the extreme positions, the noise floor is reached at lower frequencies than for the positions in the middle because there is only minor humidity gradient outside the mixing zone (see Fig. 8). For the scans along $y$, the noise floor is higher than for the scans along $x$ due to the weaker sensitivity related to shorter optical path. The estimated standard deviations due to uncorrelated noise are in the range $0.3 - 1 \cdot 10^{15}$ cm$^{-3}$ which is close to our prediction of

the random error given in sec. 4. Several distinct peaks visible at the higher end of the spectra are probably related to electrical interferences.





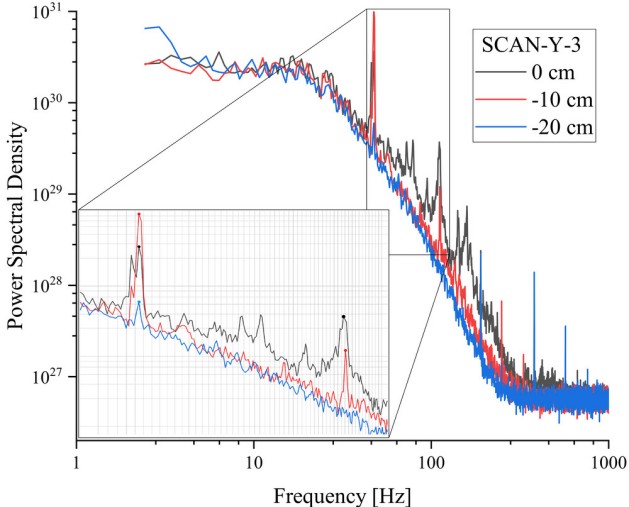

**Figure 12.** Power spectral densities of the timeseries $n(t)$ recorded at various positions $y$ during SCAN-Y-3. The insert shows the peaks in the spectrum described in the text.

## 6 Summary and discussion

We adapted FIRH, an instrument employing open-path tunable diode laser absorption spectroscopy, to perform humidity measurements in the LACIS-T wind tunnel. This application realizes a contactless optical sampling from outside the tunnel which eliminates the influence of the sensor on the investigated processes. The configuration of the setup allows for scanning at both perpendicular orientations: across the long and short dimensions of the rectangular measurement section of LACIS-T.

Three major physical factors which strongly influence the measurement were identified: self-broadening of the absorption line, interference in the glass windows and parasitic absorption in the ambient air outside the tunnel. We developed correction methods which satisfactorily account for these effects.

The accuracy of the measurement of water vapor concentration was assesed with the two approaches: a priori – taking into account the errors introduced by instrumental and external factors, and experimental – comparing FIRH with a reference hygrometer. For scanning along $x$, the expected relative and absolute errors are 7.4% and $8 \cdot 10^{15}$ cm$^{-3}$. For scanning along $y$, those errors are 8.5% and $2.3 \cdot 10^{16}$ cm$^{-3}$, respectively. The dominant contribution to the relative error comes from the inaccuracy of the absorption cross section. The dominant contribution to the absolute error results from the uncertain window transmission. The comparison between FIRH and DPM indicated that the two instruments agree well within the expected error range across the most of the investigated humidity range $n = 0.3 \dots 6.1 \cdot 10^{17}$ cm$^{-3}$ which is equivalent to $T_d = -21 \dots +21\,°\mathrm{C}$ at $T = 23\,°\mathrm{C}$. Only at low humidity ($n < 10^{17}$ cm$^{-3}$, equivalent to $T_d < -5.4\,°\mathrm{C}$) the values are overestimated by FIRH due to the decisive impact of window transmission and ambient air absorption.

The turbulent mixing of the two air streams differing in temperature and humidity was studied with FIRH and DPM for a few settings of the tunnel. The profiles of mean $n$ across the mixing zone measured with the two instruments exhibit similar





behavior, however there is a systematic offset between them. We attributed it to the limited accuracy of FIRH, the displacement of the DPM inlet with respect to the FIRH optical path and the inherent difference in sampling regimes relevant for those instruments. Those factors gain particular importance in the environment of strong humidity gradient. In the experiments where conditions allowed for the mixture of the two streams to become close to saturation or even reach supersaturation, water

vapor was observed to condense on the DPM inlet and cause malfunctions of this instrument. The contactless measurement with FIRH is not affected by such an issue as long as there is no condensate already suspended in the air.

    Thanks to the high temporal resolution of FIRH ($\sim$2 kHz), we analyzed the turbulent fluctuations in the mixing zone. The variance maximizes in the central part which coincides with the strongest gradient. It is higher for larger differences of initial $n$ between the two input streams. The width of the mixing zone is $\sim$5 cm, being in agreement with temperature fluctuation

studies performed by Niedermeier et al. (2020). Further experiments are desired to explain the mechanisms responsible for the modes identified in the autocorrelation functions and power spectra for the sampling positions close to the central plane. The cloud formation studies at LACIS-T are unaffected by these modes because the configuration for cloud measurements involves the enabled aerosol flow which provides a homogeneous mean velocity profile in the central part (see Niedermeier et al. (2020)).

Contactless optical sampling of high temporal resolution provided new insights into the properties of turbulence and turbulent mixing in LACIS-T. The results on humidity fluctuations complement the previous characterizations of turbulent velocity and temperature fields (Niedermeier et al., 2020). Nonetheless, the interpretation of FIRH measurements in the context of the processes studied at LACIS-T is not straightforward because it yields the values averaged over the length of the optical path in contrast to the localized measurements of velocity or temperature (e.g. with hot- and cold- wire devices).

Flexible contactless sampling was achieved at the cost of non-negligible parasitic absorption and window transmission effects. These factors limit the accuracy and complicate the measurement and data evaluation procedures. It would be desirable to reduce their influence in future application, e.g. with anti-reflective coatings or the integration of emitter and detector into the windows.

    The inherent limitation for the application of FIRH is the requirement of stationary conditions because in the present config-

uration the records for the absorbing $\lambda_M$ and reference $\lambda_R$ wavelengths need to be collected consecutively. LACIS-T ensures such stationarity, however this might not be the case for other laboratory facilities or field measurements. Another advancement which would come along with simultaneous differential sampling is the capability for a reliable measurement of air humidity despite cloud droplets present in the optical path. Currently, we are working on improvements to overcome this limitation and examining the signatures of droplets penetrating the optical path.

*Data availability.* The measurement records collected within this study are available from the authors upon request.



*Author contributions.* JLN, DN, TS, SPM and FS designed the study. TS, JM and RG adapted and prepared the FIRH instrument for the application at LACIS-T. JLN, DN, TS, JM, RG, LO, SS, JV and WF performed the measurements. RG and JLN processed and analyzed the collected data with advice from TS and SPM. JLN, TS and RG wrote the manuscript with contributions from SPM and DN. All authors critically proof-read and revised the manuscript.

*Competing interests.* The authors declare they have no competing interests.

*Acknowledgements.* This project has received funding from the European Union's Horizon 2020 research and innovation programme through the EUROCHAMP-2020 Infrastructure Activity under grant agreement No 730997. The development of FIRH instrument was supported by Polish National Science Center (NCN) under grant agreement No 2016/23/B/ST7/03441.



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
