# Peer review of "Contactless optical hygrometry in LACIS-T"

_Atmospheric Measurement Techniques, 2022_

## Author Comment (AC1)

**Authors' Response to the Anonymous Referee #1**

Jakub L. Nowak, Robert Grosz, Wiebke Frey, Dennis Niedermeier, Jędrzej Mijas, Szymon P. Malinowski, Linda Ort, Silvio Schmalfuß, Frank Stratmann, Jens Voigtländer, Tadeusz Stacewicz

We are grateful to the Referee #1 for the insightful comments and suggestions on our manuscript. We respond to them in detail below. The original review is given in black, our anwers in blue.

**General remarks**

I really enjoyed reading the first part of the paper. I had the feeling that all the questions risen during reading were answered in the subsequent sentences or paragraphs. Unfortunately, this impression was failed at Section 5 and 6. In my opinion, there was a break in the flow of the manuscript. The presentation of the results, and particularly its discussion remained non-conclusive. In the end I could not tell why the measurement was conducted for, and why was it important to carry out the measurements in a turbulent flow. How this measurement helps in such applications? I hoped that this question will be addressed in Summary and Discussion, but it was not the case. Anyhow, as I mentioned, the topic is very important, and the results are interesting and promising, but I wish a more detailed discussion with respect to the application in a turbulent flow.

We formulated the aim of the measurements in the introduction before:

The goal of the series of experiments was two-fold: (1) to evaluate the properties of FIRH under a wide range of well-defined reproducible conditions resembling those in the real atmosphere, (2) to characterize the humidity field and turbulent fluctuations of humidity inside LACIS-T for different settings of the wind tunnel.

The key point is that previous cloud-formation studies conducted at the LACIS-T facility (Niedermeier et al., 2020) included the measurements of droplet spectra, velocity fluctuations and temperature fluctuations but did not include the measurements of humidity fluctuations. Therefore, our work complements previous efforts with an important additional piece of information.

Following the Referee's comment, we added a paragraph at the beginning of section 5 which reminds the second goal of the study and explains the purpose of the measurements series in a more clear manner:

In this section, we intend to reach our second goal formulated at the beginning: characterize the humidity field and turbulent fluctuations of humidity inside LACIS-T for different settings of the wind tunnel. The previous cloud formation studies conducted at this facility included the measurements of droplet spectra as well as turbulent fluctuations of velocity and temperature (Niedermeier et al., 2020) but the properties of the humidity field, specifically its turbulent fluctuations could not be evaluated so far. The knowledge about these fluctuations is of

great importance for the understanding and interpretation of past and future cloud formation studies at LACIS-T. Therefore, we performed several measurement series named scans in order to investigate the mixing of the two air streams differing in thermodynamic properties. We selected the conditions which have been already used in former studies (Niedermeier et al., 2020).

Moreover, we rearranged section 6 to underline the motivation given above.

**Specific comments**

Line 33: The authors list numerous hygrometers, but in my opinion an important type of instrument is missing, namely a
photoacoustic based hygrometer. Although such a hygrometer is implicitly cited, but could also be referred here (see e.g.,
Szakall et al., Frontiers In Physics, 2020; or Tatrai et al., AMT, 2015). These papers address a lot of similar problems
as the hygrometer of the present manuscript has, like antireflection coating, and multiple reflection from windows, for
instance.

We agree that photoacoustic spectroscopy is one of the key measurement methods in hygrometry. We supplemented the overview of current hygrometers with the suggested references (Tátrai et al., 2015; Szakáll et al., 2020).

2. Fig.1, and Fig 3: Probably that was my fault, but it was for me very difficult to figure out what is x direction and what is y direction. The caption in Fig. 3. did not help either ("x position – long path, perpendicular to what is depicted in this scheme"; does not tell for me anything). Then I found the description in line 351 which helped a lot: "across the long and short dimensions of the rectangular measurement section of LACIS-T". (Probably it was written earlier, but I have overseen it?) Please consider showing x and y directions in Figure 1. Further, in caption of Fig. 1 please indicate what DPM means.

We refined our terminology to "sampling across long/short dimensions" and changed the acronyms denoting the experiments to COMP-L, COMP-S, SCAN-L, SCAN-S where L and S refers to long and short dimension, respectively. This convention is now explained in sec. 2.2. A "scan" is defined in sec. 5 as a series of measurements performed across long dimension at different positions x (SCAN-L) or across the short dimension at different positions y (SCAN-S). We corrected the captions of Fig. 1 and 3 accordingly and added the axes of the coordinate system in Fig. 1.

- Line 97: Please revise: "one can calculate water vapor concentration" I found the word "easily" superfluous.
   We removed the word "easily".
- 4. Line 132: Why did you use an electrooptic amplitude modulator? Semiconductor lasers can be easily modulated with their currents. Was that because of the disturbing effect of a residual wavelength modulation? Furthermore, in the Summary you mention the difficulties with measuring at two wavelengths with this setup. Would that be possible to apply

**Figure 1 corrected.** Schematic of the measurement section of LACIS-T. A and B mark the two air streams which are mixed in the measurement section. The red arrow marks the location where aerosol particles can be injected. Axes are included in order to display the geometry where z = 0 is the tip of the aerosol inlet, and x = 0 and y = 0 are the centerlines of the two transverse dimensions of the measurement section. The red lines denote the position of the Fast InfraRed Hygrometer (FIRH) optical paths. The thick grey lines denote the inlet tubing of the dew point mirror (DPM) hygrometer. Adapted from Niedermeier et al. (2020).

wavelength-modulation instead of amplitude modulation, and to apply 1f or 2f detection? That would also eliminate the problem with the window signal, I suppose.

We did not modulate the laser light intensity with the laser current because manipulating the current introduces changes in wavelength and, in consequence, in absorption cross section. Our measurement strategy accounts for the dependence of absorption cross section on water vapor concentration (sec. 3.1.) due to self-broadening which would be more challenging to achieve with wavelength modulation applied.

We applied slow wavelength variation when analysing the influence of the windows (sec. 3.2.). The amplitude modulation applied in actual humidity measurements served for the purpose of reducing signal noise. We consider 1f or 2f detection as a direction of possible improvements of our setup where the goal is to measure humidity fluctuations in the presence of cloud droplets.

5. Line 192: Are the two windows here the two opposite windows in the setup, i.e. in LACIS-T?

**Yes. We clarified this sentence.**

6. Caption Figure 5: The assumed concentration given here is the water vapor concentration in LACIS-T?

It is one example value of water vapor concentration in LACIS-T selected from the range of n considered in this study (see Table 1). We used such water vapor concentration in experiments COMP-L and COMP-S.

7. Line 200: I understand that the windows were large, so any antireflection coating or tilting would not work. But the laser spot is small, so not the whole window should be tilted or coated.

We agree that a fixed coating of certain spots on the glass windows would work. However, it would reduce the scanning flexibility to these spots and thus the universal purpose of the wind tunnel. Furthermore, it should be mentioned that FIRH is not a fixed instrument at LACIS-T, i.e., the wind tunnel windows are not customized for this particular instrument and the associated wavelengths. FIRH was used during a measurement campaign to evaluate the properties of FIRH and to characterize the humidity field and turbulent fluctuations of humidity inside LACIS-T.

Following the experience related to the influence of interference in the windows gained in the course of this study, an alternative method of reducing interference fringes by wavelength modulation was designed by Winkowski and Stacewicz (2021). We intend to apply this method in future experiments with FIRH.

8. Line 215: What does "perpendicular orientation" here mean?

Perpendicular to the one which is discussed in the previous sentences and shown in Fig. 6, i.e. across the short dimension of the wind tunnel. We clarified this in the text.

9. Line 220: The effects of reflection are discussed. Would such a reflection not worsen the laser efficiency when coming back to the active material of the laser? Or is this somehow avoided?

This effect is not relevant for our setup because there is no coupling between the reflected beam and the fiber. We used single mode fibers and the lens couplers with a very small angle of acceptance. Then the coupling is not possible without special adjustments.

10. Line 230: Why is the parasitic absorption so different for the x and y directions?

Absorption  $\mathcal{A} = 1 - \mathcal{T}$  depends on cross section  $\sigma$ , concentration n and optical path L. Parasitic absorption is the same for the two directions in terms of absolute values because optical path outside the wind tunnel is  $L_l = 5.0 \pm 0.3$  cm for both directions. However, in line 230 we consider parasitic absorption in relation to the absorption inside the wind tunnel which is larger for longer optical path inside (80 cm vs 20 cm). We rephrased this sentence to avoid confusing the value of parasitic absorption with the ratio of parasitic absorption to the absorption in the wind tunnel.

11. Line 242. Please consider providing the formula (maybe in the Appendix). It could be interesting for the readers or other researchers with similar applications.

As explained in sec. 4, we neglected the dependence  $\sigma(n)$ . Then Eq. (5) provides a direct formula for n:

$$n = \frac{1}{(\sigma_M - \sigma_R)L} \left[ \ln \left( \frac{I_1(\lambda_M) \mathcal{T}^{(g)}(\lambda_M)}{I_2(\lambda_M)} \frac{I_2(\lambda_R)}{\mathcal{T}^{(g)}(\lambda_R) I_1(\lambda_R)} \right) - (\sigma_M - \sigma_R) n_l L_l \right]$$

We used a common linearized approximation based on total derivative for a function of many variables  $n = n(x_1, \ldots, x_i, \ldots, x_m)$

$$\Delta n \approx \sum_{i} \left| \frac{\partial n}{\partial x_{i}} \right| \Delta x_{i}$$

which applied to the above formula and assuming

$$\begin{array}{lcl} \frac{\Delta I}{I} & = & \frac{\Delta I_1(\lambda_M)}{I_1(\lambda_M)} = \frac{\Delta I_2(\lambda_M)}{I_2(\lambda_M)} = \frac{\Delta I_1(\lambda_R)}{I_1(\lambda_R)} = \frac{\Delta I_2(\lambda_R)}{I_2(\lambda_R)} \\ \frac{\Delta \sigma}{\sigma} & = & \frac{\Delta \sigma_M}{\sigma_M} = \frac{\Delta \sigma_R}{\sigma_M} \\ \sigma_M & \gg & \sigma_R \end{array}$$

can be simplified to a form

$$\Delta n = \left(2\frac{\Delta\sigma}{\sigma} + \frac{\Delta L}{L}\right)n + \frac{1}{\sigma_M L}\left(4\frac{\Delta I}{I} + \frac{\Delta\mathcal{T}^{(g)}(\lambda_M)}{\mathcal{T}^{(g)}(\lambda_M)} + \frac{\Delta\mathcal{T}^{(g)}(\lambda_R)}{\mathcal{T}^{(g)}(\lambda_R)}\right) + n_l \frac{L_l}{L}\left(\frac{\Delta n_l}{n_l} + \frac{\Delta L_l}{L_l}\right)$$

where as  $\sigma_M$  we plugged a fixed value corresponding to a typical water vapor concentration in LACIS-T (5·1017 cm-3). The term proportional to n in the r.h.s. is what we called relative error, the remaining part of the r.h.s. was called absolute error.

The term related to water vapor concentration in the lab can be analogously obtained from Eq. (1) and Clausius-Clapeyron equation:

$$\frac{\Delta n_l}{n_l} = \frac{\Delta T_l}{T_l} + \frac{M_v L_v}{R T_l} \frac{\Delta T_{dl}}{T_{dl}}.$$

where  $M_v$  is molar mass of water,  $L_v$  is latent heat of vaporization.

We shortly explained the above method in sec. 4., however refrained from presenting the entire derivation as it is pretty straightforward once the method is known.

12. Line 254: Here the measurement was conducted with two air streams. If I understood correctly, the former measurements were carried out without flow. The measurement conditions should be described correctly and at the beginning of the paragraphs. Here it is also not clear how the sampling for the dew point mirror was done. Or was the inlet permanently in LACIS-T, as shown in Figure 1?

The measurements of window transmission described in sec. 3.2 were carried out without the flow. It is specified in line 203. The subsequent measurement series were COMP-L (formerly called COMP-X, see point 2 in this response) and COMP-S (formerly called COMP-Y) in which indeed the two air streams were used, however with the same velocity, temperature and humidity in both. The dew point mirror inlet was permanently inside LACIS-T as in Fig. 1, however its horizontal position was changed so that it is always beneath (i.e. downstream of) the optical path of FIRH. This was specified in lines 93 and 270. We clarified the description of DPM sampling in sec. 2.1 and 4 as well as of the measurement conditions in sec. 4.

13. End of Section 4: For me the explicit determination of the detection limit or the minimum detectable concentration of FIRH is missing. From the calibration it could be determined, right? Something like 1.5 E17 cm-3.

We preferred not to specify the exact detection limit as it depends on the optical path length and such estimation would need to rely on the dew-point mirror measurements. Instead, we provide a range of water vapor concentration  $1.0...6.1 \cdot 10^{17} \text{ cm}^{-3}$  for which we verified the agreement between FIRH and DPM.

14. Line 283: Again the question: was the DPM inlet permanently mounted? Or was that movable? One could perform a scan with DPM if its inlet is movable.

The DPM inlet was permanently inside LACIS-T but its position was changed in accordance with FIRH so that the inlet was always beneath (i.e. downstream of) the optical path. This was specified in line 93. The scans were performed simultaneously with the two instruments FIRH and DPM. The positions of both were adjusted manually which might have caused some inaccuracies.

15. Line 286: That FIRH measurements represent an average along the optical path is not a new information, it is mentioned a few lines earlier.

We removed the repetition.

16. Line 295: "The profiles of n ... " – was already mentioned.

We removed the repetition.

17. Line 317: Is it possible to measure the air flow and get information about the velocity profile? Applying an LDV, for instance?

The air-flow was measured independently with a Hot-Wire anemometer and the results are given in Niedermeier et al. (2020) as already stated in the text. LDV would also be possible, however, it requires the insertion of seed particles as tracers. Currently, particle insertion is only possible via the aerosol inlet. This would mean that these velocity measurements would be limited to the locations where the particles are. Those particles might also influence the humidity field (through water adsorption/absorption) which we want to avoid here.

18. Line 333, 335: Vibration and oscillation of the window are the same thing, if I understand correctly. Why were the windows vibrating? Some mechanical vibration from the whole facility?

Yes, we refined to one term: vibrations. The windows vibrate to a minor extent due to the mechanical vibrations of the whole facility. For example, thermostats are used for the adjustment of the air-flow temperature. These thermostats cause vibrations that are damped by the design of the wind tunnel, but are still transferred to the measurement section and thus to the glass windows.

19. Figure 12: The inlet figure has no scale, so it is difficult to understand it.

We added the scale to the insert in Fig. 12.

Figure 12 corrected. Power spectral densities of the timeseries n(t) recorded at various positions y during SCAN-S-3. The insert shows the peaks in the spectrum described in the text.

- 20. Lines 372-375: I did not understand the motivation of this discussion. Are the results meaningful in this aspect or not? We are confident with the conclusions given in lines 372-375. We suppose the Reviewer might have meant lines 375-379 which indeed contained some discussion that, we agree, is unnecessary in this section. We removed those sentences.
- 21. Line 380: It is claimed here that the measurements "provided new insights into the properties of turbulence and turbulent mixing in LACIS-T". This is not obvious for me and that is what I meant in my General remarks.

We removed this sentence as superfluous. The next one explains what we meant by new insights, namely that the results on humidity fluctuations complement the previous characterizations of turbulent velocity and temperature fields from Niedermeier et al. (2020). Following the general remarks given by the Reviewer, we included a new paragraph at the beginning of sec. 5. to clarify the usage of these investigations and also rearranged sec. 6.

22. Data availability: I suggest the authors using a data repository for publishing the data, at least the ones corresponding to the figures.

We prepared a dataset corresponding to the figures and will reference it in the final version of the manuscript.

**References**

Niedermeier, D., Voigtländer, J., Schmalfuß, S., Busch, D., Schumacher, J., Shaw, R. A., and Stratmann, F.: Characterization and first results from LACIS-T: A moist-air wind tunnel to study aerosol-cloud-turbulence interactions, Atmospheric Measurement Techniques, 13, 2015– 2033, https://doi.org/10.5194/AMT-13-2015-2020, 2020.

- Szakáll, M., Mohácsi, Á., Tátrai, D., Szabó, A., Huszár, H., Ajtai, T., Szabó, G., and Bozóki, Z.: Twenty Years of Airborne Water Vapor and Total Water Measurements of a Diode Laser Based Photoacoustic Instruments, Frontiers in Physics, 8, 384, https://doi.org/10.3389/FPHY.2020.00384/BIBTEX, 2020.
- Tátrai, D., Bozóki, Z., Smit, H., Rolf, C., Spelten, N., Krämer, M., Filges, A., Gerbig, C., Gulyás, G., and Szabó, G.: Dual-channel photoacoustic hygrometer for airborne measurements: Background, calibration, laboratory and in-flight intercomparison tests, Atmospheric Measurement Techniques, 8, 33–42, https://doi.org/10.5194/AMT-8-33-2015, 2015.
- Winkowski, M. and Stacewicz, T.: Optical interference suppression using wavelength modulation, Optics Communications, 480, 126464, https://doi.org/10.1016/J.OPTCOM.2020.126464, 2021.

---

## Author Comment (AC2)

**Authors' Response to the Anonymous Referee #2**

Jakub L. Nowak, Robert Grosz, Wiebke Frey, Dennis Niedermeier, Jędrzej Mijas, Szymon P. Malinowski, Linda Ort, Silvio Schmalfuß, Frank Stratmann, Jens Voigtländer, Tadeusz Stacewicz

We are grateful to the Referee #2 for the insightful comments and suggestions on our manuscript. We respond to them in detail below. The original review is given in black, our anwers in blue.

**Comments**

1. In the description of the LACIS-T facility (Figure 1 and supporting text), it would be helpful to clarify the geometry by showing axes x, y and z and indicate the reference positions for each (z = 0 being the tip of the aerosol inlet, and x = 0 and y = 0 being the centerlines of the two transverse dimensions of the duct) and define the longitudinal (z) position of the FIRH sampling and discuss why that position was chosen (I see in L142 "at the height 39 cm downstream [of] the place where the two streams merge"—meaning z = 39 cm downstream of the aerosol inlet?). It would also help to mention that the duct is oriented vertically (it is, right?) so that describing the position of the DPM sampling inlet as "below" (= displaced in z, downstream of) the FIRH beam makes sense (I was originally picturing the inlet offset in y for x "scans" and in x for y "scans"). In Figure 1, I'm not sure what information I am supposed to extract from the picture/diagram to the right of the two (x and y view) schematics of the LACIS-T measurement section, and it seems like it should be omitted or discussed.

   We agree with the suggestions and corrected the description of the facility and the measurement setup accordingly. We added the information about the vertical orientation of the measurement section of LACIS-T in sec. 2.1. We clarified the geometry by adding axes in Fig. 1. and expanding its caption.

   Following also the remarks of Referee #1, we refined our terminology to "sampling across long/short dimension" and changed the acronyms denoting the experiments to COMP-L, COMP-S, SCAN-L, SCAN-S where L and S refers to long and short dimension, respectively. This convention is now explained in sec. 2.2:

   > The sampling of the air inside LACIS-T was achieved across the glass windows at the height $z = 39$ cm, i.e. downstream of the aerosol inlet where $z$ is the longitudinal position with $z = 0$ being the tip of the aerosol inlet, see Fig. 1. This height was selected because previous measurements related to cloud formation studies were perfomed at the same position by Niedermeier et al. (2020). The emitter and the photodetector PD2 were mounted on a rigid aluminium sleigh at the opposite sides of the wind tunnel (see Fig. 3) as close to the glass windows as it was possible (while maintaining the flexibility of easy changes of the scanning position) in

order to minimize the optical path outside the wind tunnel. Nevertheless, even despite drying the ambient air in the laboratory, parasitic absorption could not be entirely avoided (see sec. 3.3). The sleigh enables scanning the spatial variability of humidity statistics by moving the sensor horizontally along the walls of the wind tunnel. Two separate sleighs were prepared to allow measurement at both transverse orientations: across the long ($L_L = 80 \pm 0.3$ cm) and short ($L_S = 20 \pm 0.3$ cm) dimensions of the rectangular measurement section of LACIS-T, denoted hereafter with letters L and S, respectively. The sampling across the long dimension was possible at the positions $x = -3.25 \ldots 2.75$ cm due to the thickness of the window frame. In the case of the sampling across the short dimension, the positions $y = 0, -10, -20$ cm were selected in this study. The coordinates $x$ and $y$ denote two transverse dimensions with the origin of the coordinate system located in the center of the measurement section as shown in Fig. 3.

[Figure]

**Figure 1 corrected.** Schematic of the measurement section of LACIS-T. A and B mark the two air streams which are mixed in the measurement section. The red arrow marks the location where aerosol particles can be injected. Axes are included in order to display the geometry where $z = 0$ is the tip of the aerosol inlet, and $x = 0$ and $y = 0$ are the centerlines of the two transverse dimensions of the measurement section. The red lines denote the position of the Fast InfraRed Hygrometer (FIRH) optical paths. The thick grey lines denote the inlet tubing of the dew point mirror (DPM) hygrometer. Adapted from Niedermeier et al. (2020).

2. L111: "the exact tuning...prevents interferences" isn't quite correct. If one or both of the wavelengths were near an absorption line from another molecule, the measurement would be impacted regardless of how exact (precise) the tuning. It is the choice of the specific H2O absorption feature that is sufficiently far from interfering absorption lines that is important.

We agree. We meant that the exact tuning in addition to the proper choice of the absorption feature ensures selectivity of the measurement. We specified this in the text.

3. L148: "transverse" might be a better word than "perpendicular", and specify relative to the direction of flow.

We agree and changed the word.

4. L149 (and Figure 3): similar to above, it would be helpful to have the origin of the coordinates defined and the range of possible values (-x … +x, -y … +y).

The origin is in the center of a measurement section which is indicated in Fig. 3. The ranges of the used $x$ and $y$ positions were added to the text. They are also listed in Table 1.

5. L151: since the two wavelengths are achieved by adjusting only the laser diode current, why are the two measurements made in separate (long) periods instead of quasi-simultaneously by rapid tuning between the two?

Currently, our in-house developed software does not feature rapid tuning between the two wavelengths with desired accuracy and precision which are important for determination of the exact absorption cross sections. We consider wavelength modulation between the two extremes as a direction of possible improvements of our setup where the goal is to measure humidity fluctuations in the presence of cloud droplets.

6. Figure 5: I think "interferred" is meant to be "interfered", but that is not used as an adjective. I think the appropriate term would be "convolved" as the product of the convolution of the absorption and fringe spectra.

The label in the figure was changed into "convolved".

7. Figure 6: it would be nice to add lines indicating the locations of M and R.

The lines indicating $\lambda_M$ and $\lambda_R$ were added to the figure.

8. L218 (and L265): why is the value of Tx(g)(lambda) = 0.87 so much lower than Ty = 0.98?

The net transmission depends on the exact thickness of the glass and the exact distance between the two windows in relation to the wavelength which is of the order of 1 μm only. As discussed in lines 192-197, the range of possible values can be $0.748 \leq \mathcal{T}_2 \leq 1$. All the four derived $\mathcal{T}^{(g)}$ fit into this range. It is rather a coincidence that $\mathcal{T}_x^{(g)}(\lambda_M)$, $\mathcal{T}_y^{(g)}(\lambda_M)$, $\mathcal{T}_y^{(g)}(\lambda_R)$ are so close to 1 while $\mathcal{T}_x^{(g)}(\lambda_R)$ is not.

9. L264: the systematically high values of n from FIRH would indicate that the determinations of the contributions to the signal from the windows and ambient air were low when applied to the experimental arrangement. Or the DPM was systematically low at low H2O. Did the DPM-measured value agree with the expected based on the generated H2O in the flows?

It is difficult to discern the true reason for this systematic difference between the instruments at low humidities. Indeed, if we consider the DPM as ground truth, then the window transmission term $\ln\left(\frac{\mathcal{T}^{(g)}(\lambda_M)}{\mathcal{T}^{(g)}(\lambda_R)}\right)$ is too high or/and ambient

air absorption term $(\sigma_M - \sigma_R)n_l L_l$ is too low (see Eq. (5)) but also other factors may play a role, such as considerable uncertainties of absorption cross sections $\sigma_M$, $\sigma_R$ or optical path lengths $L$, $L_l$. Following the experience gained in this study, we intend to further minimize parasitic absorption and window interference in future applications.

For the experiments COMP-L and COMP-S (formerly called COMP-X and COMP-Y), both airflows A and B were conditioned identically, i.e., they had the same $T_d$ and it was changed from measurement to measurement identically in both streams. However, there is a difference in the method of humidity control between $T_d > 0°$C and $T_d < 0°$C (Niedermeier et al., 2020). For $T_d > 0°$C, the airflows were led completely through the Nafion humidifiers, so the generated H2O amount is known very precisely. This is further monitored with two additional DPMs (not described in this study, one for each stream A and B). Generated H2O amount agrees well with the DPM used in this study located further downstream inside the measurement section. For $T_d < 0°$C, the desired humidity is achieved by mixing dry and humidified air. For each stream, the airflow is split into two parts. One part is left dry while the other part is led through the Nafion humidifier. Then, those two parts are mixed. The resulting water vapor amount can be calculated knowing $T_d$ of the dry and humidified air as well as the flow rates of those two parts (being left dry and being humidified). Such calculated $T_d$ is checked by the two monitoring DPMs. However, as the $T_d$ of the dry part is not exactly constant, we observe a slight variability in the resulting $T_d$ of the mixed airflow.

10. Section 5.1: I'm not sure that the arguments presented here really presents a complete argument explaining the systematically higher values measured by the DPM than FIRH (L284). The values are typically at a mean concentration ($> 2$e17) at which the prior experiments demonstrated good agreement, and anyway, at low values the prior experiments would predict that FIRH would be higher than the DPM measurement. Spatial differences (average vs point) would seem to require a crossover at some position since gradients along the y direction cannot explain it given the statement in L276. Given conservation of H2O in the flow, the small difference in z of the FIRH and DPM measurements shouldn't produce a significant difference (would require a source of H2O).

In the experiments of the type SCAN (sec. 5.), the conditions were significantly different than in the earlier experiments of the type COMP (sec. 4.). During COMPs, temperature and humidity was equal in the two streams A and B. During SCANs, temperature and humidity differed between the streams which allowed for the observation of the mixing profiles presented in Fig. 8.

We are not certain what is the main reason of the systematic discrepancy between FIRH and DPM observed in SCANs. We suppose all 3 factors originally given in the text can contribute. The second point was expanded to mention also possible angular misalignment of the FIRH optical path with respect to the desired $x = const$, $z = const$ plane. The third point was reformulated to mention the effective filtering related to the two sampling regimes relevant for FIRH and DPM. FIRH involves spatial filtering but provides high temporal resolution. DPM collects air from a relatively small, yet finite volume. The diameter of the inlet tube is 6 mm which is quite significant size in relation to the gradient of the order of $10^{17}$ cm$^{-4}$ and the range of investigated $x$ positions. Moreover, due to the particular measurement method,

which requires cooling a mirror to a dew-point temperature, in the environment of rapid humidity fluctuations DPM acts as a low-pass filter with rather complex and potentially non-intuitive transfer properties. The corrected discussion reads:

> The mean $n$ exhibits a significant systematic offset (shift) between FIRH and DPM in all four experiments. Several factors could contribute to the observed offset: (1) the limited accuracy of FIRH (see sec. 4), (2) displacements and misalignments between the FIRH optical path and the DPM inlet (i.e. inaccuracy in setting $x$ position, angular deviation of the FIRH path from the desired direction in the plane $x = const, z = const$, deliberate shift in $z$ between the sensors), (3) difference in sampling regime between the instruments (in fact FIRH involves spatial low-pass filtering, i.e. averaging along the optical path but provides high temporal resolution while DPM involves temporal low-pass filtering of complex characteristics but collects air from a relatively small volume). The offset is higher than observed in the comparison experiments COMP-L and COMP-S, likely due to the significant spatial gradient of humidity (up to $2 \cdot 10^{17}$ cm$^{-4}$). Such gradient was absent in those comparison experiments but here, due to the factors (2) and (3), it affects the outcome.

In future applications of FIRH, we intend to minimize the influence of window transmission and parasitic absorption which would reduce the number of factors involved and possibly allow to discern the main factor responsible for the discrepancies.

11. Figure 8: it would be interesting to compare the variance with dn/dx to graphically demonstrate the statement in L296/7 of the coincidence of the peak in variance with the steepness of the gradient.

    Calculated gradient dn/dx is shown in the modified Fig. 8.

12. L315: the hypothesis here could have been tested by comparing with an experiment including a flow (aerosol-free) from the aerosol inlet with n = (nA + nB)/2 that would be more representative of the typical aerosol-inclusive studies at LACIS-T.

    We agree with this idea. Unfortunately, at the time of the experiments we were not aware of such a strong importance of aerosol flow on humidity field but quantified this issue in the course of later data analysis.

13. Data availability: per the AMT data policy, it is (at the least) encouraged that authors make the supporting data publicly available via some repository.

    We prepared a dataset corresponding to the figures and will reference it in the final version of the manuscript.

**References**

Niedermeier, D., Voigtländer, J., Schmalfuß, S., Busch, D., Schumacher, J., Shaw, R. A., and Stratmann, F.: Characterization and first results from LACIS-T: A moist-air wind tunnel to study aerosol-cloud-turbulence interactions, Atmospheric Measurement Techniques, 13, 2015–2033, https://doi.org/10.5194/AMT-13-2015-2020, 2020.

[Figure]

**Figure 8 corrected.** Turbulent mixing of the two streams differing in thermodynamic properties (given in Table 1) observed in the course of the four experiments: the profiles of mean $n$, its variance and gradient $dn/dx$ with respect to the position $x$.